# Identification of Multiple Domains of *Entamoeba histolytica* Intermediate Subunit Lectin-1 with Hemolytic and Cytotoxic Activities

**DOI:** 10.3390/ijms23147700

**Published:** 2022-07-12

**Authors:** Kentaro Kato, Hiroshi Tachibana

**Affiliations:** 1Department of Eco-Epidemiology, Institute of Tropical Medicine (NEKKEN), Nagasaki University, 1-12-4 Sakamoto, Nagasaki 852-8523, Nagasaki, Japan; 2School of Tropical Medicine and Global Health, Nagasaki University, 1-12-4 Sakamoto, Nagasaki 852-8523, Nagasaki, Japan; 3Department of Parasitology, Tokai University School of Medicine, 143 Shimokasuya, Isehara 259-1193, Kanagawa, Japan; htachiba@is.icc.u-tokai.ac.jp

**Keywords:** *Entamoeba histolytica*, intermediate subunit lectin-1, galactose/*N*-acetylgalactosamine (GalNAc)-inhibitable lectin, hemolytic activity, cytotoxicity

## Abstract

Galactose and *N*-acetyl-D-galactosamine-inhibitable lectin of *Entamoeba histolytica* have roles in the pathogenicity of intestinal amoebiasis. Igl1, the intermediate subunit lectin-1 of *E. histolytica*, has been shown to have both hemolytic and cytotoxic activities that reside in the C-terminus of the protein. To identify the amino acid regions responsible for these activities, recombinant proteins were prepared and used in hemolytic and cytotoxic assays. The results revealed that Igl1 has multiple domains with hemolytic and cytotoxic activities and that amino acids 787-846, 968-1028 and 1029-1088 are involved in these activities. The hemolytic activities of the fragments were partly inhibited by mannose, galactose and *N*-acetylgalactosamine, and glucose showed lower or negligible inhibitory effects for the activities. This is the first report of a protozoan protein with hemolytic and cytotoxic activities in multiple domains.

## 1. Introduction

*Entamoeba histolytica* (*E. histolytica*) causes amoebiasis and an estimated 50 million cases of dysentery, colitis and extraintestinal abscesses, resulting in 40,000 to 100,000 deaths annually [1]. Galactose (Gal)- and *N*-acetyl-D-galactosamine (GalNAc)-inhibitable lectins are required for adherence of *E. histolytica* trophozoites to colonic mucins and host cells [2,3]. These lectins consist of a 260 kDa heterodimer of transmembrane heavy subunit (Hgl) and glycosylphosphatidylinositol (GPI)-anchored light subunit (Lgl), and intermediate subunit (Igl) glycoproteins. Igl is non-covalently associated with the Hgl/Lgl dimer in lipid raft-like domains and contributes to adherence of the parasite to target host cells [4,5,6,7].

There are two isoforms of Igl, referred to as Igl1 and Igl2, and both contain multiple CXXC motifs with different localization in *E. histolytica* trophozoites [8,9]. Previously, we reported the hemolytic activity of both Igl proteins and the cytotoxic activity of Igl1, which reside in the C-terminus of the protein (Igl1_603-1088_) [10,11]. The hemolytic activity of *Igl1* gene-silenced *E. histolytica* strains was lower than that of a control strain, indicating that the protein has roles in both the adherence and virulence of *E. histolytica* [11].

Development of inhibitors of these activities requires more detailed identification on the responsible amino acid regions of Igl. To achieve this goal, we prepared recombinant fragment proteins based on the Igl1 protein sequence and used these proteins in hemolytic and cytotoxic assays. Following identification of fragments with hemolytic activities, we evaluated inhibition of these fragment activities by monosaccharides in vitro.

## 2. Results

### 2.1. E. histolytica Igl1 Has Hemolytic Activity between Amino Acids 787 and 846

We have previously shown that Igl1 has hemolytic activity at its C-terminus (Igl1_603-1088_) and that three fragments (Igl1_603-846_, Igl1_726-967_ and Igl1_847-1088_) showed this activity [10]. In the present study, a further six fragments (Igl1_726-846_, Igl1_847-967_, Igl1_787-906_, Igl1_726-786_, Igl1_757-817_ and Igl1_787-846_) were generated based on the amino acid sequence of Igl1_726-967_ to narrow down the region with activity. We did not include fragments from Igl1_603-726_ in the study because Igl1_294-753_ does not have activity [10]. After purifying the recombinant proteins using a Ni-NTA column (Figure 1A), the proteins (1 μM) were incubated with horse red blood cells (HoRBCs) for several hours (Figure 1B). Among the fragments, Igl1_726-846_, Igl1_787-906_ and Igl1_787-846_ showed hemolytic activity visually, and this was confirmed by the concentration of hemoglobin (Hb) released in the supernatant of incubated samples (Figure 1B,C). These results suggest that the minimum fragment with activity in this region is Igl1_787-846_ (60 amino acids) (Figure 1D). 

### 2.2. E. histolytica Igl1 Has Other Regions with Hemolytic Activity 

The results in Figure 1 indicate that Igl1_787-846_ has hemolytic activity, but this does not explain the similar activity of Igl1_847-1088_ [10]. To confirm that Igl1 has multiple regions with this activity, we generated three Igl1 fragments (Igl1_968-1088_, Igl1_968-1028_ and Igl1_1029-1088_) based on the sequence of Igl1_847-1088_. Fragments of Igl1_847-967_ were not included because this region has no activity (Figure 1). As shown in Figure 2, all fragments studied had hemolytic activity, indicating that multiple regions in Igl1_968-1028_ and Igl1_1029-1088_ have this activity (Figure 2D). Taken together, the results in Figure 1 and Figure 2 suggest that Igl1 has at least three regions with hemolytic activity.

### 2.3. Partial Inhibition of Hemolytic Activity of Igl1 Fragment Proteins by Monosaccharides

The hemolytic activity of Igl1_14-1088_ (1 μM) is not inhibited by mixing with ten-fold higher concentrations (10 μM) of galactose or mannose [10]. In a study of the effects of glucose, lactose and GalNAc on hemolysis by a lectin-hemolytic peptide conjugate, 4_3_-CEL-I, the range of carbohydrate concentrations was 1.56–50 mM, and 25 mM GalNAc and 50 mM lactose inhibited the hemolytic activity of the conjugate by about 90% [12]. Therefore, we examined whether higher concentrations of monosaccharides could inhibit the hemolytic activities of Igl1 fragment proteins (Figure 3). The maximal inhibition of the activity of Igl1_14-1088_ was 57% with 50 mM GalNAc and 64% with 50 mM galactose, but only 37% with 50 mM mannose and 33% with 50 mM glucose (Figure 3A). The hemolytic activities of Igl1_787-846_ and Igl1_968-1028_ were inhibited by about 55% by 50 mM GalNAc, 45-50% by 50 mM galactose, and about 38% by 25 mM mannose (Figure 3B,C), and that of Igl1_1029-1088_ was inhibited by about 60% by 50 mM GalNAc, 50% by 50 mM galactose and 40% by 25 mM mannose (Figure 3D). Incubation with 50 mM glucose inhibited about 20% of the activity of Igl1_968-1028_ but did not inhibit Igl1_787-846_ and Igl1_1029-1088_ (Figure 3B–D). GalNAc treatment showed the steepest inhibition against the hemolytic activity of all Igl1 fragments at a concentration < 25 mM, but this inhibition was still only about 60% at maximum.

### 2.4. Cytotoxicity of Recombinant Fragment Proteins of Igl1 against Caco-2 Cells

We have previously shown that Igl1 has both hemolytic and cytotoxic activities [10]. To examine whether the three Igl1 fragments (Igl1_787-846_, Igl1_968-1028_ and Igl1_1029-1088_) with hemolytic activity also have cytotoxicity, Caco-2 cells were incubated with these fragments for 12 or 24 h (Figure 4), and with Igl1_726-786_, a fragment of 61 amino acids that did not show hemolytic activity (Figure 1). A cobblestone appearance of Caco-2 cells was observed after 12 or 24 h of incubation with PBST, Igl1_726-786_ or medium, but this appearance was destroyed when the cells were incubated with Igl1_787-846_, Igl1_968-1028_ or Igl1_1029-1088_ (Figure 4B). There was a significant decrease in the number of cells remaining on the plate in incubation with Igl1_787-846_, Igl1_968-1028_ or Igl1_1029-1088_, compared to that with PBST (Figure 4C). In contrast, Igl1_726-786_ showed no cytotoxicity. These results indicate that Igl1 fragments with hemolytic activity also have cytotoxicity.

## 3. Discussion

We have shown previously that the Igl subunit of *E. histolytica lectin* has hemolytic activity [10,11]. However, all the tested C-terminus fragment Igl proteins had this activity, and we were unable to identify the specific region(s) responsible for the activity. In the current study, we prepared smaller recombinant proteins and used these fragments in a hemolytic assay to clarify the regions with activity. Three fragments (Igl1_787-846_, Igl1_968-1028_ and Igl1_1029-1088_) showed activity, indicating that Igl1 has multiple regions (at least three) that contribute to hemolytic activity. This explains why a monoclonal antibody recognizing the C-terminus of Igl1 could not completely inhibit the hemolytic activity of the protein [10]. 

Monosaccharides only partially inhibited the hemolytic activities of the three Igl1 fragment proteins, even at millimolar concentrations of mannose, galactose and GalNAc, and glucose showed lower or negligible inhibitory effects. These results agree with previous findings showing that the hemolytic activity of Igl1_14-1088_ could not be inhibited by 10 μM galactose or mannose [10] and that 250 mM glucose could not inhibit Igl1 binding to Chinese hamster ovary (CHO) cells [7]. In contrast, intact Igl protein can be purified from *E. histolytica* using a Gal-affinity column [7], and the protein binds to GalNAc-BSA neoglycoprotein-coated beads [13]. These results indicate that Igl has a strong affinity for galactose and GalNAc, and it has also been shown that Igl binding to CHO cells is inhibited by 250 mM galactose or GalNAc [7]. Thus, the lectin domain(s) of Igl1 may not have a major role in hemolytic activity. However, Igl lacks the carbohydrate recognition domain (CRD) found in other lectins, and identification of this domain is required for proving or disproving this possibility. Using glycoproteins with more complex carbohydrates in an inhibition assay is also required for further studies.

Cytotoxic activities were also observed for Igl1_787-846_, Igl1_968-1028_ and Igl1_1029-1088_, but not for Igl1_726-786_, which has a similar molecular size to those of the three other fragments. Igl1_818-846_ would be a fragment with activity, and we tried to obtain recombinant proteins with 30 amino acids, but it was difficult to generate these proteins in *E. coli* culture. Recombinant proteins with hemolytic activity also had cytotoxicity, suggesting that there are common molecular mechanisms underlying these activities. These mechanisms were not examined in this study, but this is the first report to show that a protozoan protein can have both hemolytic and cytotoxic activities in multiple regions. 

The results in the previous [10] and the present studies suggest that Igl1_787-846_, Igl1_968-1028_ and Igl1_1029-1088_ have fewer hemolytic and cytotoxic activities than Igl1_14-1088_. This indicates that the three regions are required for the full activities of Igl1, but we need further studies to prove this speculation. 

Two of the three regions in Igl1 with these activities (Igl1_968-1028_ and Igl1_1029-1088_) are close to a GPI-anchored domain of the protein, as illustrated in Figure 5A. Therefore, on the plasma membrane of *E. histolytica*, the regions might be hindered intramolecularly or by associated subunits of the lectin, such as Hgl and Lgl (Figure 5B) [5]. However, gene suppression of Igl1 in *E. histolytica* resulted in less hemolytic activity of the gene-suppressed strain, indicating that the protein is exposed and has a role in the activity on site [11]. The reactivity of sera from patients with amoebiasis, including asymptomatic cyst passers, is highest against Igl1_603-1088_ [14], and anti-Igl1_603-1088_ antibodies in sera have effects against amoebic liver abscess formation in hamsters [15]. These findings also indicate that this region of Igl1 is exposed on the surface of *E. histolytica*. However, it is unclear whether Igl1 anchored to the cell membrane has hemolytic and cytotoxic activities or if Igl1 needs to be cleaved or shed from the membrane to permit these activities. 

There is no common amino acid sequence among Igl1_787-846_, Igl1_968-1028_ and Igl1_1029-1088_, as shown in Figure 6A. Comparison with the sequences of Igl1_726-786_ or Igl1_757-817_, which did not have hemolytic activity, also failed to answer why some Igl1 fragments have hemolytic and cytotoxic activities (Figure 6B). The top 20 amino-acid motifs in hemolytic peptides have recently been predicted using neural networks [16], but none of these motifs were found in the hemolytic regions of Igl1. Single proline substitutions in the *Bordetella pertussis* CyaA pore-forming fragment reduced its hemolytic activity [17]. In contrast, a proline residue at position 14 of *Trichoderma* alamethicin is essential for the hemolytic activity [18]. It is possible that the number of prolines in the fragments affects the structure and activity of Igl1, since there are one or two prolines in all Igl1 fragments with activity, but four or five in the fragments without activity. 

Hydrophobicity correlates to the hemolytic activity of antimicrobial peptides [19]. Igl1 fragments with hemolytic activity have hydrophobic amino acids in the N-terminus around amino acid position 18 and hydrophilic regions in the C-terminus (Figure 7). This characteristic may confer hemolytic and cytotoxic activities to the fragments. However, a pore-forming *E. histolytica* amoebapore-A which has hemolytic activity does not have the characteristic. In any case, structure analyses of Igl1 protein are required to obtain a clearer answer. Collectively, the results of this study suggest the need to identify inhibitors of all regions of Igl1 with hemolytic activity to reduce the virulence of *E. histolytica* related to the protein.

## 4. Materials and Methods

### 4.1. Expression and Refolding of Recombinant Igl1 Proteins and Ni Column Purification

Recombinant proteins with a His-tag at the N-terminus were expressed in *Escherichia coli* ECOS^TM^ competent BL21(DE3) cells (Nippon Gene Co., LTD., Toyama, Japan) using pET19b expression vector (69677-3, Novagen, MA, USA) and the primers shown in Table 1. The proteins were further purified using a Ni column, as described in detail elsewhere [10,11,14]. 

### 4.2. SDS-PAGE and Coomassie Brilliant Blue Staining of Purified Recombinant Proteins

Recombinant proteins or bovine serum albumin (1 µg each) were mixed with SDS sample buffer (Invitrogen, CA, USA) and subjected to SDS-PAGE. The gel was treated with SimplyBlue Safe stain solution (Invitrogen, CA, USA) and incubated until blue bands appeared on the gel [10,11].

### 4.3. Hemolytic Assays Using Recombinant Lectins and Measurement of Released Hemoglobin

Hemolytic assays and quantification of hemolytic activity were conducted as previously described [10]. Briefly, recombinant Igl1 proteins (2 µM each, 50 µL) in PBST were mixed with 50 µL 2% (*v*/*v*) horse red blood cells (HoRBCs) (Japan Bio Serum, Tokyo, Japan) in PBS at room temperature, and images were taken at several time points. A Hemoglobin B Test Kit (Wako, Osaka, Japan) was used to measure the concentration of hemoglobin (Hb) in supernatants of RBCs incubated with recombinant proteins for 6 h. The results are expressed as the mean of 5 experiments with a standard deviation (SD). For inhibition assays using D(+)-galactose (Cat. No. 071–00032), D(+)-mannose (Cat. No. 130–00872), *N*-acetyl-D-galactosamine (Cat. No. 013–12821) and D(+)-glucose (Cat. No. 049–31165), monosaccharides were included in the incubations at final concentrations of 50 mM to 1.56 mM using 2-times serial dilutions. All monosaccharides were purchased from Fujifilm Wako Pure Chemical Corporation, Japan. The Hb concentration in the supernatant of the samples without monosaccharides was defined as 100% hemolytic activity for data analysis. The results are shown as the mean of 3 independent experiments with SD.

### 4.4. Cytotoxicity Assay

Caco-2 cells (ATCC, HTB-37) were cultured in MEM basic medium (Gibco, Beijing, China) supplemented with Earle’s salts, L-glutamine and 20% fetal bovine serum. After detachment with 0.25% Trypsin-EDTA (Gibco, NY, USA), the cells were cultured in a 96-well plate at approximately 2 × 10^4^ cells/100 µL/well at 37 °C under 5% CO_2_ for 24 h. Volumes (100 µL) of 2 µM recombinant Igl1 proteins, PBST or medium containing 200 units/mL penicillin G and 200 µg/mL streptomycin (Wako, Osaka, Japan) were added (0 h), and the cells were incubated for an additional 12 or 24 h under the same conditions. Images of Caco-2 cells in a 96-well plate were taken with an EVOS-XL microscope. The cells were trypsinized and harvested after 0, 12 or 24 h of incubation, and the number of cells per well was counted. The results are shown as the mean of 5 experiments with SD.

### 4.5. Statistical Analysis

Multiple comparisons were performed by ANOVA with a Dunn test or Dunnett test, with *p* < 0.05 considered to be significant.

## Figures and Tables

**Figure 1 ijms-23-07700-f001:**
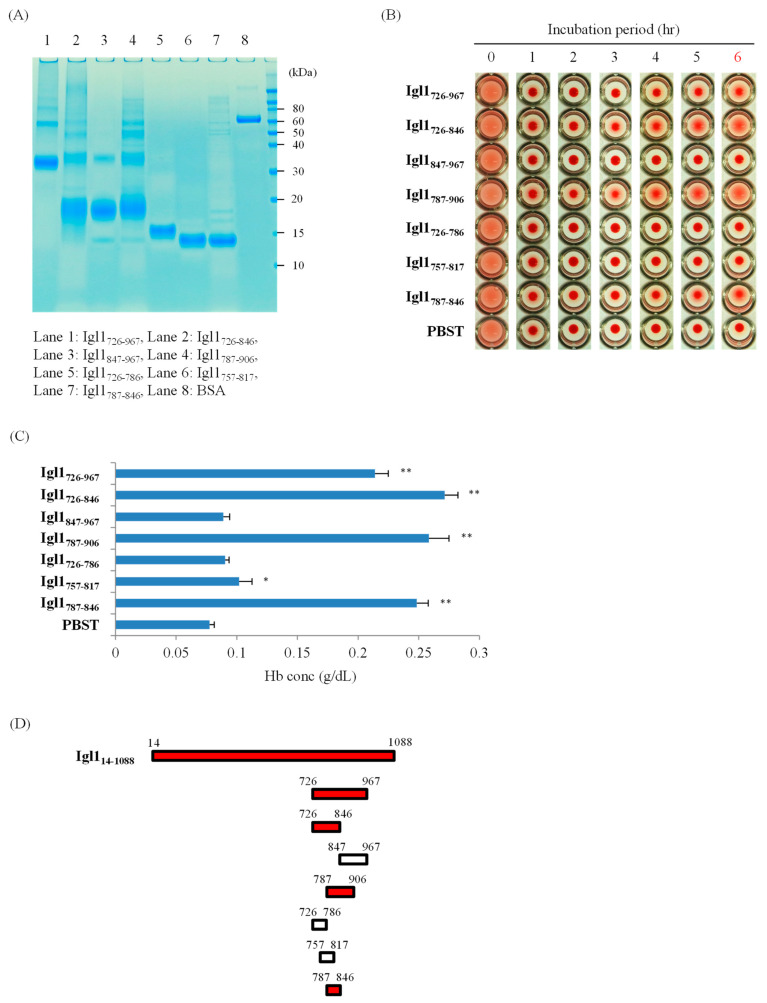
Hemolytic activity of recombinant Igl1 fragment proteins based on Igl1_726-967_. (**A**) Purities and amounts of recombinant proteins were confirmed by SDS-PAGE, with 1 µg of each protein run on the gels. (**B**) HoRBCs were incubated in a 96-well plate with Igl1 fragment proteins for the indicated periods. Representative images of 5 independent studies are shown. (**C**) Concentrations of hemoglobin (Hb) released in the supernatant of samples incubated for 6 h. Data are the mean + SD from 5 independent experiments. Hemolytic activities of Igl1 fragments shown in bars with asterisks were significantly higher than those of phosphate-buffered saline with 0.05% Tween 20 (PBST) (* *p* < 0.05, ** *p* < 0.01 by ANOVA with a Dunn test). (**D**) Summary of data for fragment proteins with hemolytic activities. Recombinant Igl1 fragments with hemolytic activity are shown in red bars. Open bars indicate fragment proteins without this activity.

**Figure 2 ijms-23-07700-f002:**
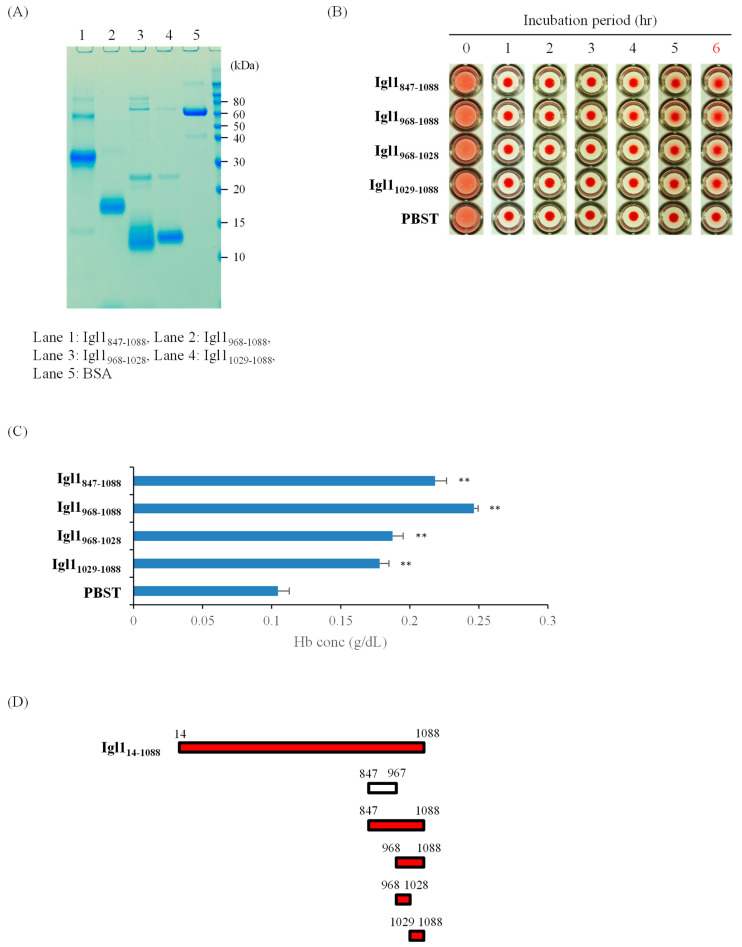
Hemolytic activity of recombinant Igl1 fragment proteins based on Igl1_847-1088_. (**A**) Purities and amounts of recombinant proteins were confirmed by SDS-PAGE, with 1 µg of each protein run on the gels. (**B**) HoRBCs were incubated with Igl1 fragment proteins in a 96-well plate for the indicated periods. Representative images of 5 independent studies are shown. (**C**) Concentrations of hemoglobin (Hb) released in the supernatant of samples incubated for 6 h. Data are the mean + SD from 5 independent experiments. Hemolytic activities of Igl1 fragments shown in bars with asterisks were significantly higher than that of PBST (** *p* < 0.01 by ANOVA with a Dunn test). (**D**) Summary of data for fragment proteins with hemolytic activities. Recombinant Igl1 fragments with hemolytic activity are shown in red bars. The open bar (Igl1_847-967_) indicates a fragment without activity (Figure 1).

**Figure 3 ijms-23-07700-f003:**
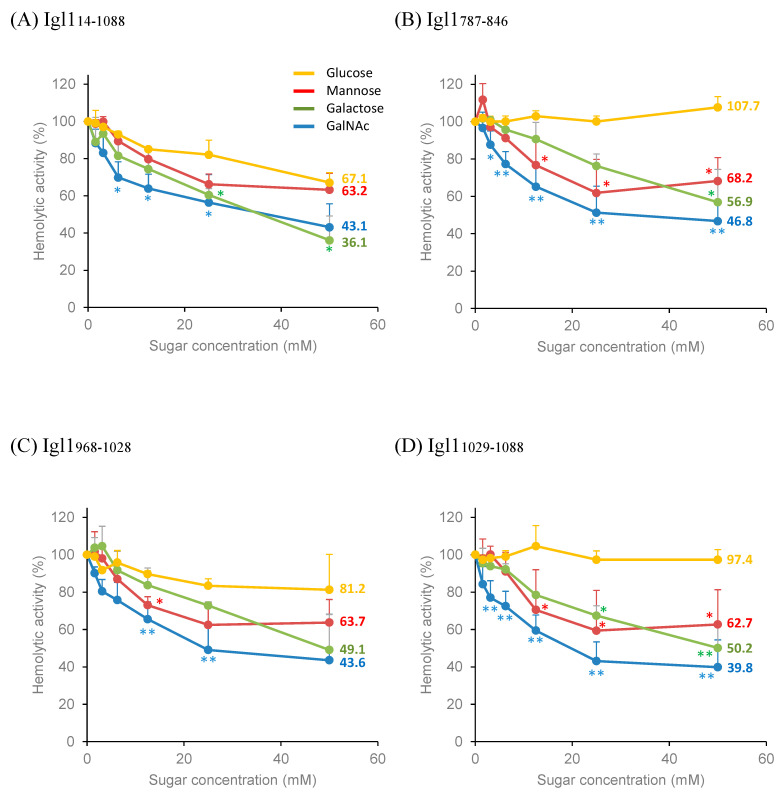
Effects of monosaccharides on hemolytic activities of Igl1 fragment proteins. Igl1 fragments and 2% (*v*/*v*) horse red blood cells were incubated with or without monosaccharides for 6 h at room temperature. Hb concentrations in the supernatant were measured and those in samples without monosaccharides were defined as 100% hemolytic activity. Numbers in the line graphs indicate the hemolytic activity (%) of Igl1 fragments treated with 50 mM monosaccharides. (**A**) Igl1_14-1088_. (**B**) Igl1_787-846_. (**C**) Igl1_968-1028_. (**D**) Igl1_1029-1088_. Data are the mean + SD from 3 independent studies. Asterisks indicate hemolytic activities of Igl1 fragments with mannose, galactose or GalNAc that are significantly lower than that with glucose (* *p* < 0.05, ** *p* < 0.01 by Dunnett test).

**Figure 4 ijms-23-07700-f004:**
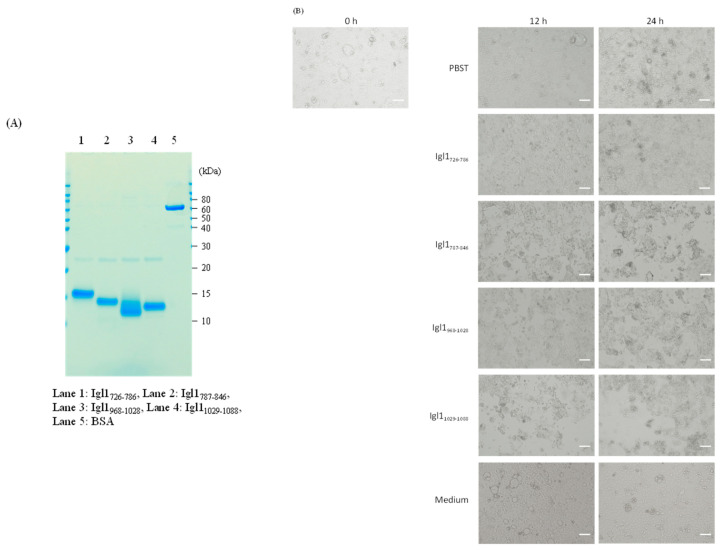
Cytotoxicity of Igl1 fragment proteins against Caco-2 cells. Caco-2 cells were cultured with 1 μM Igl1 fragments and incubated at 37 °C for 12 or 24 h. (**A**) Purities and amounts of Igl1 fragments were confirmed by SDS-PAGE using a NuPAGE gel, with 1 µg of each protein run in the gel. Lane 1: Igl1_726-786_, Lane 2: Igl1_787-846_, Lane 3: Igl1_968-1028_, Lane 4: Igl1_1029-1088_, Lane 5: BSA. (**B**) Images of Caco-2 cells incubated with Igl1 fragments in a 96-well plate for the indicated periods. Images were taken with an EVOS-XL microscope. Bars indicate 100 μm. (**C**) Number of Caco-2 cells remaining in the well after incubations for 0, 12 and 24 h. Data are the mean + SD from 5 independent studies. ** *p* < 0.01 vs. PBST (Dunnett test).

**Figure 5 ijms-23-07700-f005:**
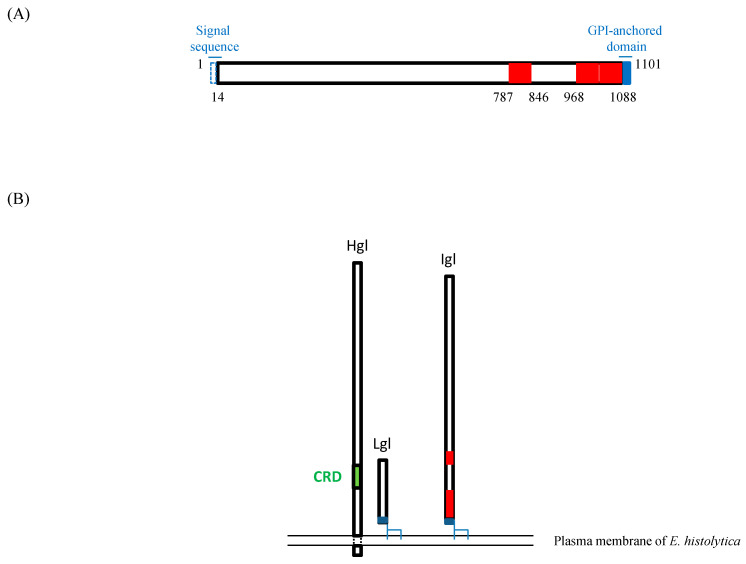
Hemolytic and cytotoxic regions of Igl1. (**A**) The Igl1_14-1088_ recombinant fragment lacks the signal sequence (aa 1–13) and GPI-anchored domain (aa 1089–1101). Regions in red are those with both hemolytic and cytotoxic activities. (**B**) Schematic of Hgl, Lgl and Igl on the cell membrane of *E. histolytica*. Igl is associated with other lectin subunits: Hgl and Lgl. CRD: carbohydrate recognition domain.

**Figure 6 ijms-23-07700-f006:**
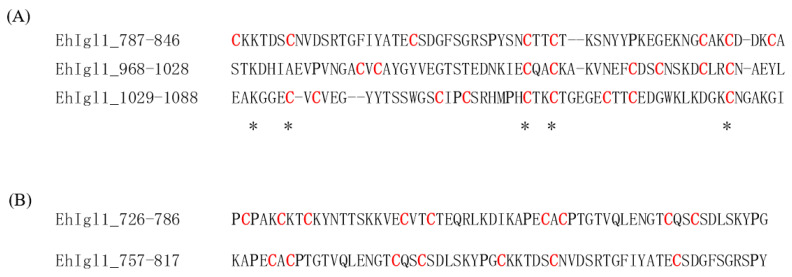
Amino acid sequences of Igl1 fragments. (**A**) Sequence alignment of Igl1 fragments with hemolytic and cytotoxic activities. Asterisks indicate common amino acids. (**B**) Sequences of Igl1 fragments without these activities. The Igl1 sequence refers to protein ID AAK92361. Cysteine residues are shown in red and proline residues are shown in bold.

**Figure 7 ijms-23-07700-f007:**
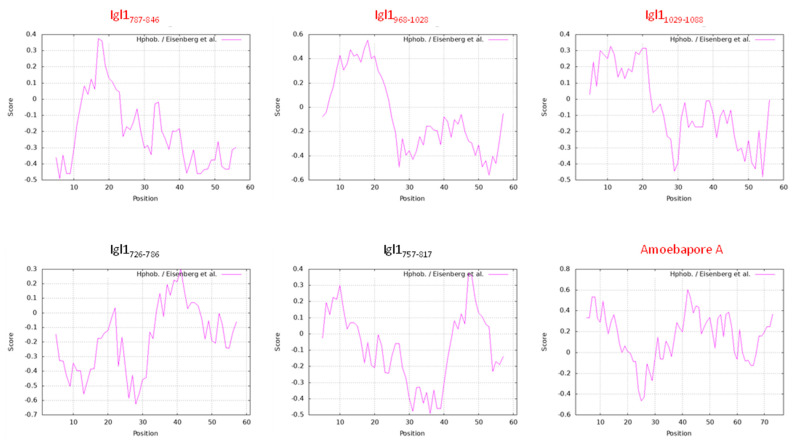
Hydrophobicity of Igl1 fragments and amoebapore-A. Hydrophobicity of Igl1 fragments and amoebapore-A (PDB: 1OF9_A) was calculated by ProtScale program in Expasy (https://web.expasy.org/protscale/) (accessed on 6 July 2022) with an amino acid scale published by Eisenberg et al. [20]. Igl1 fragments in red or black indicate the fragments with or without hemolytic activity, respectively.

**Table 1 ijms-23-07700-t001:** Oligonucleotide primers used in the study.

Primer	Position ^a^	Sequence (5’ to 3’) ^b^
EhIgl-S14	40–59	CCCTCGAGGATTATACTGCTGATAAGCT
EhIgl-S726	2176–2195	CCCTCGAGCCATGTCCTGCAAAATGTAA
EhIgl-S757	2269–2289	CCCTCGAGAAAGCACCAGAATGTGCTTGT
EhIgl-S787	2359–2378	CCCTCGAGTGTAAAAAAACTGATTCATG
EhIgl-S847	2539–2558	CCCTCGAGACATGTTCAGATAAAGACAC
EhIgl-S968	2902–2921	CCCTCGAGTCAACAAAAGATCATATTGC
EhIgl-S1029	3085–3105	CCCTCGAGGAAGCAAAAGGAGGAGAATGT
EhIgl-AS786	2341–2358	CCCTCGAGTTATCCTGGATATTTTGAAAG
EhIgl-AS817	2434–2451	CCCTCGAGTTAATAAGGACTACGTCCACT
EhIgl-AS846	2521–2538	CCCTCGAGTTATGCACATTTATCATCACA
EhIgl-AS906	2701–2718	CCCTCGAGTTACTTGTATGTATCACTCTC
EhIgl-AS967	2884–2901	CCCTCGAGTTAACATTTAGTACATTCTTC
EhIgl-AS1028	3066–3084	CCCTCGAGTTATAAATATTCAGCATTGCAT
EhIgl-AS1088	3247–3264	CCCTCGAGTTAAATGCCTTTAGCTCCATT

^a^ Nucleic acid numbering is based on the *E. histolytica* Igl1 gene sequence (AF337950). ^b^ Nucleotides added for cloning and translation termination are underlined.

## Data Availability

The data presented in this study are available in the article.

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
