# Peer review of "Identification of Multiple Domains of Entamoeba histolytica Intermediate Subunit Lectin-1 with Hemolytic and Cytotoxic Activities"

_ijms, 2022, doi:10.3390/ijms23147700_

Round 1
Reviewer 1 Report
This manuscript describes the identification of regions responsible for the hemolytic and cytotoxic activities of Igl1 from Entamoeba histolytica. The manuscript is well written, the approach is straightforward, and the conclusions are supported by the results.
There are minor issues that should be addressed:
1. define PBST
2. How the Igl1 fragments were prepared should be described in more detail. At a minimum the plasmid used should be given although it would be useful to have a brief description of the other methods used in their preparation.
3. It is unclear what the numbers in parentheses in lines 390-392 are. Are these catalog numbers?
4. The supplementary figure was not provided but from the description it might be useful to have directly in the article.
5. It would be useful to have some indication of how strong the original hemolytic and cytotoxic activities of the different fragments are compared to the full-length Igl1 (14-1088). This would provide some information as to whether the activities are additive or is it really sufficient for just one of these domains to be present to get the same activity as the full length protein. This should be added to the discussion.
Author Response
Dear Reviewer 1,
Thank you very much for your comments and suggestions.
Please see the attachment for the response to your comments.
Best regards,
Kentaro Kato

Reviewer 2 Report
1. L-20: Inhibition of hemolytic activity of those fragments by glucose was less than other sugars, but it exists for Igl114-1088 and Igl1968-1028. Authors should modify the sentence (“The hemolytic activities of the fragments ………galactose and N-acetylgalactosamine, but not by glucose.”). The abstract should also include the results obtained in Fig. 5.
2. L-60: Along with the minimum fragment with hemolytic activity, readers can be interested to know which one was the most active fragment. Fig. 1B also suggests that fragment 787-906 possesses the highest activity. Is it correct? If yes, authors should mention it.
3. L-110: Please elaborate the term ‘PBST’.
4. L-326: Authors are suggested to provide references showing the possible involvement of Prolines (or any other amino acid) to exert such hemolytic or cytotoxic activities, in case of lectins from protozoa/other microbes/invertebrates/fungi.
5. To understand the last paragraph of the 'Discussion', authors are suggested to include Fig. S1A and S1B as a new figure in the manuscript.
Author Response
Dear Reviewer 2,
Thank you very much for your comments and suggestions.
Please see the attachment for the response to your comments.
Best regards,
Kentaro Kato

Reviewer 3 Report
The manuscript is devoted to the study of an extremely interesting object, namely the amoeba toxin, which causes a number of serious pathological processes in the human body. The authors continued a more detailed study of the modular structure of toxin subunit molecules. Using shorter recombinant proteins, the authors tried to elucidate the role of each domain in the manifestation of cytotoxic, lytic, and agglutinating activities.
Unfortunately, the authors failed to achieve effective hemagglutination and its complete abolition by the studied monosaccharides; perhaps this could be achieved using glycoproteins with more complex carbohydrate structures. The broad carbohydrate specificity of agglutinins and lysins is also surprising, as is the large lag period of up to several hours. Amphiphilic molecules such as naturally occurring polyphenols can cause false-positive haemagglutination, but this possible role for the amphiphilicity of the polypeptides studied is not discussed in the manuscript. In addition, the lysis of red blood cells can be caused by the formation of pores in the membrane with the release of such a large molecule as hemoglobin.
In any case, this interesting article will be useful to many readers, as well as to young scientists. It can be accepted for publication after some comments from the authors.
Author Response
Dear Reviewer 3,
Thank you very much for your comments and suggestions.
Please see the attachment for the response to your comments.
Best regards,
Kentaro Kato
